# A Proposal of Fault Tree Analysis for Embedded Control Software

**Masakazu Takahashi [1,*]** , **Yunarso Anang [2]** and **Yoshimichi Watanabe [1]**

[1]   Department of Computer Science and Engineering, University of Yamanashi, Kofu,
     Yamanashi 400-8511, Japan; nabe@yamanashi.ac.jp
[2]   Department of Computational Statistics, Politeknik Statistika STIS, East Jakarta 13330, Indonesia;
     anang@stis.ac.id
*   Correspondence: mtakahashi@yamanashi.ac.jp; Tel.: +81-55-220-8585

**Abstract:** There are many industrial products in our life, and the actions of those products are controlled by embedded control software (ECSW). Recently, many troubles have been caused by ECSW. To avoid those troubles, it is necessary to clarify the causes of the troubles and take countermeasures. However, the results of those tasks depend on the skills of the analyst. This paper proposes an analytic method that clarifies the causes of troubles by applying fault tree analysis (FTA) to the ECSW. The characteristics of the proposed method are as follows: Preparation of fault tree templates (FTTs) corresponding to instructions of the ECSW, and definition of the FT development rules by combining FTTs according to the back-tracing of the instruction execution process. By complying with the proposed method strictly, when an analyst who has studied computer science and safety engineering for 2–3 years conducts FTA, the analyst can obtain an appropriate result of FTA. This indicates that the safety level of ECSW will improve. As a result of applying the proposed method to existing ECSWs, we find that we can obtain the result of FTA at the appropriate level.

**Keywords:** fault tree analysis; embedded control software; safety analysis; industrial products

---

## 1. Introduction

This paper proposes an analysis method of unexpected events (top events) caused in embedded control software (ECSW) for industrial products written in the C language. The proposed method realizes a safe ECSW by analyzing the causes of a top event and taking countermeasures.

Industrial products dealt with by this paper are home electronic products, medical equipment, product facilities, automobiles, aircrafts, space equipment, etc. ECSW is software that controls motions of an industrial product. We assume that the target ECSW has the following characteristics: Working on a single-chip CPU with a single-core, having several thousands of lines of code (KLOC), written in the C language, and not using newly developed technologies. The reason those assumptions are adopted is as follows: The rate of usage of the single-core CPU is about 60%, the median of lines of code (LOC) is 6.7 KLOC, the rate of usage of the C language is about 60%, and the rate of development without newly developed technologies is 80% [1]. The reason the median of LOC is used is as follows: Even the LOC of ECSW is distributed from 0.1 to 1551 KLOC, the average of the LOC is 41 KLOC, and the rate of ECSW with over 50 KLOC is under 10%. Additionally, as the large-scaled ECSW is built by combining several control equipment using a network and/or data-bus, the LOC of ECSW installed into individual equipment is nearly the same as the LOC of the target ECSW. Fault tree analysis (FTA) is a safety analysis method that identifies the causes of a top event (fundamental events) by tracing logical relationships between a top event and its causes. In FTA, the logical relationships between a top event and fundamental events are shown as tree structures called the fault tree (FT).

In the past, the completeness of the developed FTs depended on the skills and experiences of the analyst. There exists a problem where the analyst sometimes omits the events in the FT that are well known from his experiences, and other analysts sometimes cannot understand the reason why the FT is developed. By applying the proposed method, analysts are able to conduct FTA in the code level of the ECSW. Analysts who have studied computer science and safety engineering for 2–3 years will be able to conduct adequate FTA. As the proposed method defines the concrete FT development rules, even those with less experience will easily understand the FT analysis. Additionally, FTs are developed within adequate time using the developed FTA support tools (FTs with ECSW having several hundred LOC can be developed within 1 to 2 h). As a result, the safety level of the ECSW can be improved using the proposed method.

These days, accidents of industrial products resulting from a top event of ECSW have become an issue. To avoid the top event of the ECSW, countermeasures for the top event have to be taken, in addition to the conduction of appropriate design and tests. There exists an FTA as a method that identifies the fundamental events from the top event. Many FTA methods have already been proposed as an analysis method for the ECSW, but those proposals have some problems, such as the unclearness in the FTA conduction sequence, and inconsistency in the FT analytic granularity (function level and component level). Therefore, the developed FT includes some differences (oversights in the items investigated and variance in the granularity for the items investigated) depending on the experiences and skills of the analyst. As a result, there exist some cases where the results of FTA become insufficient. This paper proposes an analysis method that can develop equivalent and appropriate FT whenever an analyst conducts FTA.

Here, we shall discuss an overview of the FTA related to ECSW proposed in this paper. Before conducting FTA for ECSW, we conduct a system-level FTA for a system-level fault and clarify the hardware and software faults and parts (instruction of ECSW) that cause a system-level fault. FTA is executed for the software fault. First, we will discuss top events (Faults). The top event of the ECSW, which is the subject of the analysis, is defined as an unexpected event occurring under a certain condition: For example, functions do not start (or stop) at a designated timing, processing becomes frequent, data are not updated at the designated timing, or calculation results deviate from the permitted range. Simple programming errors should be detected in tests, and as these are carried out essentially and efficiently, such errors are not covered here. Next, we will discuss the procedure for executing FTA for ECSW. Regarding the instruction and the top event as the starting state, the instructions executed and events that occurred immediately before (middle events) are clarified. If this reverse tracking work is repeated, we can reach the fundamental events and the instructions causing them. To perform reverse tracking work, it is necessary to prepare (a) a method for clarifying the relationship between the events prior to and after the execution of each instruction, and (b) a method for determining instructions executed immediately before the instruction being executed. For (a), we prepare an FT template expressing the relationship between the events before and after executing each instruction. For (b), we define rules for identifying instructions executed immediately before instructions currently being executed. The input of the proposed method is as follows: The fault of software, the instruction that causes the fault, and the source code (besides, information on the ECSW structure extracted from ECSW is issued, details are described in Section 3.3.2). The output is the developed FTs.

The rest of this paper is organized as follows. Section 2 describes the related works. Section 3 describes the occurrence process of the top event, the development of the FT template, the definition of the FT development rules, and the prototyped FTA support tools. Section 4 describes the application and evaluation of the proposed method and the issues related to the proposed method. Finally, Section 5 describes the summary of this research.

## 2. Related Works

In this section, we provide an outline of the establishment of standards related to ECSW safety in the industrial field, the ECSW safety analysis method, and the methods to improve reliability using FTA and related methods.

First, we discuss the ECSW development standards in various industrial fields. The reason for establishing development standards is based on the idea that a high level of safety can be realized by developing ECSW in accordance with appropriate standards. Supervisory authorities for industrial products are required to have a high level of safety and have imposed a duty on the manufacturing companies to develop ECSW that conforms to the development standards. Examples of such regulations are JIS T 2304 [2], IEC 62,304 [3], and IEC 82304-1 [4] in the medical devices field; Good Automated Manufacturing Practice [5] in the pharmaceutical manufacturing field; ISO 26262 [6] in the automobile field; and DO-178C [7] and JAXA JMR-001 [8] in the aerospace field.

Next, we discuss the various types of safety analysis methods. Takahashi et al. proposed a method to clarify the faults that might happen in ECWS for pharmaceutical manufacturing devices, as well as to clarify the countermeasure comprehensively using Failure Mode and Effects Analysis (FMEA) [9]. Weber et al. analyzed the causes of faults using FTA for ECSW in aerospace, which is written in assembly language [10]. Thapaliya et al. proposed an FTA method for the preliminary design written in Unified Modeling Language (UML), which is a representative object-oriented modeling language [11]. Leveson et al. prepared a fault tree (FT) Template that clarifies the causes of faults for each fundamental software instruction and showed that an FT can be developed related to faults by combining these [12,13]. However, as the method of combining the FT templates to develop an FT is not defined, the level of completeness of the FT is dependent on the ability of the analyst. Therefore, Takahashi et al. proposed rules for combining FT templates and developing an FT while reverse-tracking the process in which faults occur [14]. Park et al., with regard to a nuclear power plant control system developed using a Function Block Diagram with a Programmable Logic Controller, proposed a method in which the FT is developed by preparing and combining FT templates in relation to the Function Block [15]. Hansen et al. demonstrated a method in which a Hazard and Operability Study (HAZOP) is applied to the software written in UML [16]. Hulin et al. defined HAZOP guide words used when applying HAZOP to software and applied these to software in which a high level of safety is required [17]. Weiss applied System Theoretic Process Analysis (STPA), which is a hazard analysis method resulting from interactions between system components, to ECSW of an earth orbiting satellite [18]. Furthermore, Takahashi et al. have proposed a method in which STPA is applied to embedded software described in UML, and software hazards are clarified [19]. Although such methods have contributed to the improvement in ECSW safety, they have not been put into widespread use.

Finally, we discuss the applications of FTA and related methods. Kloos et al. proposed a test case development and execution method for the top event of safety-critical ECSW and showed the improvement in the test coverage [20]. Chen et al. proposed the FT development method. The method develops the state machine-type failure model using EAST-ADL and develops the FT by applying the Hip-HOPS [21], where EAST-ADL is a modeling language for the automotive embedded system, and Hip-HOPS is a dependency analysis method. Trawczynski et al. proposed a method for discovering the problem that cannot be found using the classical method [22]. The method randomly injects faults into the Distributed Brake System, evaluates the effects by the injected faults to the system, and identifies the faults. Dabboussi et al. proposed a method to develop FTs related to the top event of the vehicular ad hoc network (VANET). This method develops reliability block diagrams of VANET equipment, creates the FT from the diagram configuration, and calculates the reliability of the VANET using fault data (mean time between failure, availability, etc.) for equipment that is included in the VANET [23].

## 3. Proposed FTA Method

In this section, we describe the proposed method, the FT template, the FT development rules, and the FT support tool, which are described in Sections 3.1–3.4, respectively.

### 3.1. Overview of the Proposed Method

In this subsection, we provide an overview of the ECSW top event occurrence process, the FT Template, the FT development rules, and the proposed method.

#### 3.1.1. Execution Process of the Top Event

First, we explain the top event of ECSW. In general, when conducting FTA for an embedded system, the system-level FTA for the fault is conducted, and the faults (the top event) for hardware and software are clarified. The system-level FTA is shown in Figure 2. The proposed method is applied to the software fault. The software fault is considered as follows: The value of the specific variable is too large or small, and the function cannot invoke or cannot terminate, etc.

Next, we explain the occurrence process of the top event. We describe the process by which top events occur in the ECSW. First, we assume the ECSW operates normally until a certain instruction is executed. Then, after the next instruction is executed, it is assumed that a slight deviation occurs from the normal state and a top event occurs when executing the n-th instruction. Here, the j-th executed instruction is described as $I_j$, and the j-th occurring event as $<Event_j>$. The sequence of instructions and events from the time at which the fundamental event occurs to the time at which the top event occurs can be described as follows. This is referred to as an execution process. The execution process includes a return from interruption, branches by a condition branch instruction, and confluences by a repeat instruction. For example, the execution process of $I_{101}$ in Figure 1 shows a return from interruption, and $<Event2_{201}>$ shows the branch by the condition branch instruction.

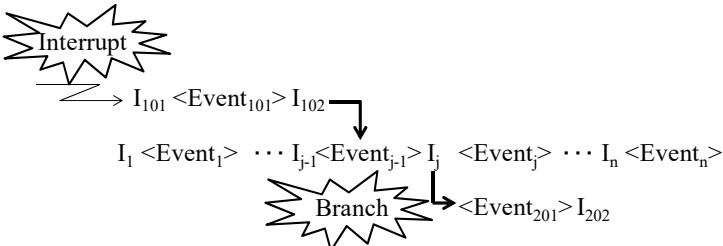

**Figure 1.** Example of an execution process.

#### 3.1.2. FT Templates and FT Development Rules

Here, we describe the FT Template and FT development rules. With $<Event_n>$ and $I_n$ as the starting point of Figure 1, if we perform reverse-tracking on the execution process, we can reach $<Event_1>$ and $I_1$. Therefore, a method for determining $I_{j-1}$, which is a precondition for executing $I_j$, is defined as the FT Development Rules (FDRs). Additionally, the relationship between the change from $<Event_j>$ to $<Event_{j-1}>$ before and after executing $I_j$ is defined as the FT Template (FTT). Hereafter, from the perspective of simplifying the understanding of the explanation of the algorithm, the top event is defined as $<Event_1>$, the instruction generating this as $I_1$, the fundamental event as $<Event_n>$, and the instruction generating this as $I_n$ by numbering the index in reverse order. The suffix j in $I_j$ and $<Event_j>$ is used to identify them uniquely and does not mean the order of execution.

#### 3.1.3. Outline of the Proposed Method

To conduct FTA for the ECSW using the proposed method, it is necessary to decide the top event and instruction that causes the top event. There are two types of top events. One is the top event that is caused by the ECSW itself; the other is the top event that is caused by the combination of the hardware's action and ECSW's behavior under specific conditions. The former events are as follows: The overflow and underflow of the variable, and non-formation of the conditions for starting and stopping the function. The instructions that cause such top events are as follows: The instruction that calculates the value of the variable, and the instruction that executes and terminates

the specific function. The latter events are not needed, as the ECWS's behavior itself is inappropriate. By conducting the system-level safety analysis (FMEA and/or STPA, etc.) for the industrial product, it is necessary to clarify the combination of both the hardware's action and ECSW's behavior under the specific condition (the system-level safety analysis is beyond this research). These top events are as follows: The value of the variable exceeds the threshold, and it either invokes or does not invoke the specific function under the specific conditions. Those instructions that cause those top events are the same as the instructions such that the ECSW itself causes the top event. In the following explanation, it is assumed that the ECSW's top event and instructions have already been identified.

Figure 2 shows an overview of the proposed method that consists of the preparation stage and the development stage. In the preparation stage, we analyze the existing ECSW, and develop a FTT for instructions used with a high degree of frequency (described later in Section 3.2). In addition, we define the FDR (described later in Section 3.3). The FT corresponding to the top event is developed during the development stage. First, the top event and the instruction that causes the top event are defined by conducting system-level safety analysis. Next, pre-processing is performed in the ECSW to be analyzed. During pre-processing, instructions for which FTT has not been prepared are replaced with instructions for which the equivalent FTT has been prepared. Third, the pre-processed ECSW is analyzed, and the ECSW information required for FTA is extracted. Finally, based on the last pre-processed ECSW, the top event, the ECSW information, the FTT, and the FDR are used to develop the FT. The reason the pre-processing is performed is that it is difficult to prepare FTT for all instructions in the C language in the view of cost. When new frequently used instructions appear, the FTTs for these instructions are developed and added. As FTT only stipulates the relationship between the events before and after executing the instruction, there is no impact on existing FTTs and the FDR.

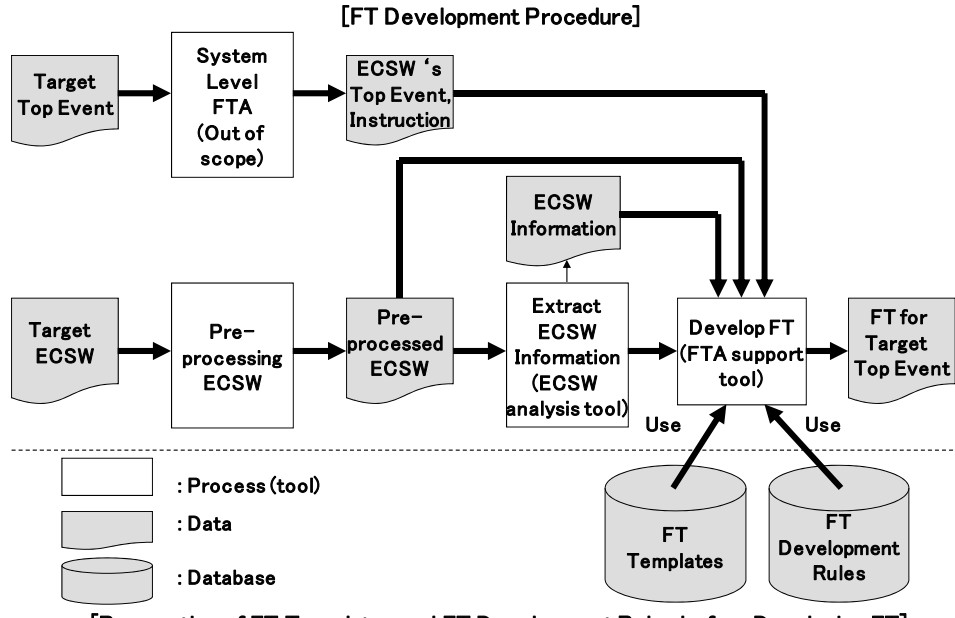

**Figure 2.** Outline of the proposed method.

### 3.2. FT Templates

In this subsection, we discuss FTTs. In this research, we analyze the existing ECSW, and prepare FTTs for nine types of frequently used instructions. The reason we analyze the existing ECSW is that the instructions frequently used in FTA and their FTT can be revealed because the ECSW does not use the newly developed technologies. Additionally, as the FTT shows only the relationship between events before and after the execution of the instructions, the addition of new FTTs does not affect other existing FTTs.

In Figure 3, we show the symbols used in FTT. As the symbols used for the supplementary explanation are only shown in the notes, these do not affect the meaning of the FT.

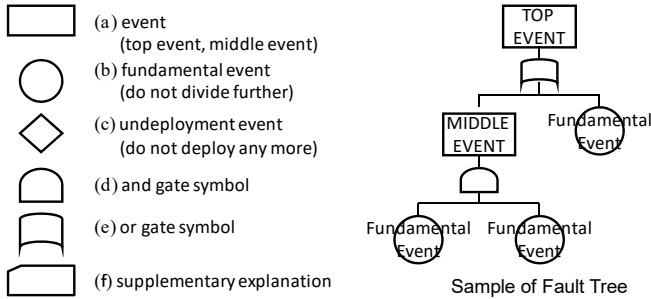

**Figure 3.** Fault tree (FT) symbols used in fault tree template (FTT).

### 3.2.1. FTT for Assignment Statement

Figure 4 shows the FTT for the assignment (variable = expression (value)). This FTT indicates that an assignment causes an event when the substituted value generates an event, or the operator generates an event. This FTT is the same as the FTT in [12].

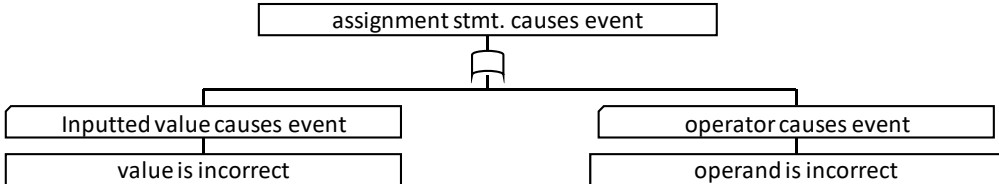

**Figure 4.** FTT for assign statement.

### 3.2.2. FTT for Block If Statement

Figure 5 shows the FTT for block if statement (if (condition1) {statement} else if (condition2) {statement} - - - else {statement}). This FTT indicates that the block if statement generates an event in the case one or more events occur within the condition node. Furthermore, an event occurs in the case "the i-th condition node is established" and "the i-th process causes an event," or in the cases the "else node is established" and the "else node processing causes an event."

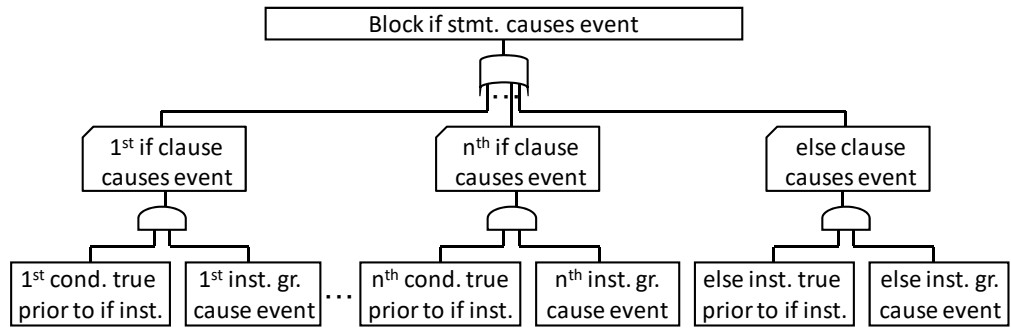

**Figure 5.** FTT for block if statement.

### 3.2.3. FTT for While Statement

Figure 6 shows the FTT for the while statement (while (condition) {statement}). This FTT shows that the while statement causes an event in the case "the event occurs because the while statement is not executed," or "the event occurs because the while statement is executed." The former expresses cases in which the while statement is not executed while the repeated conditions are not established,

whereas the latter shows that the repeated conditions are established, and the event occurs because the statement execution is repeated n-times (including the repeat limit in a while loop). This FTT is similar to the FTT in [12].

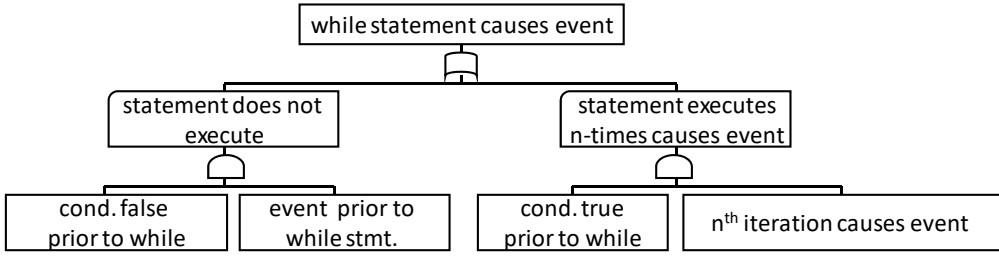

**Figure 6.** FTT for while statement.

### 3.2.4. FTT for Function Call

Figure 7 shows the FTT for the function call statement. This FTT expresses the fact that the function call statement generates an event in the case the function parameters generate an event, or the function is not appropriately started. This FTT is similar to the FFT in [12].

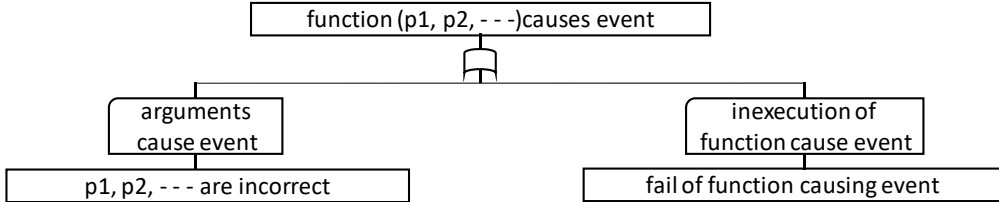

**Figure 7.** FTT for function call.

### 3.2.5. FTT for Interrupt

Figure 8 shows the FTT for the interrupt. This FTT expresses the fact that an interrupt generates an event in the case "the event occurs when the interrupt occurs," or "the event occurs because the interrupt does not occur." The former refers to the case where the interrupt is generated, and as the interrupt routine is executed, the event occurs. The latter can be divided into cases where the interrupt is not generated and cases where it is disabled. In cases where the interrupt is not generated, this expresses cases in which the interrupt does not occur, and because the interrupt routine is not executed, the event occurs. Cases where the interrupt is disabled express cases where the event occurs because the interrupt is disabled (prohibited).

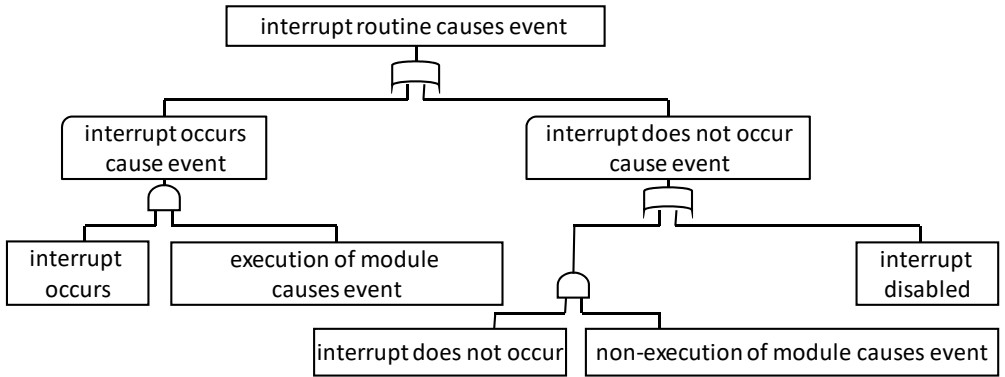

**Figure 8.** FTT for interrupt.

### 3.2.6. FTT for Global Variables

Figure 9 shows the FTT for the global variables. Global variables can exist anywhere within the ECSW. This FTT expresses the fact that in cases where a global variable causes the event, there is a relationship with one or more instructions in which the used global variable is set for the value.

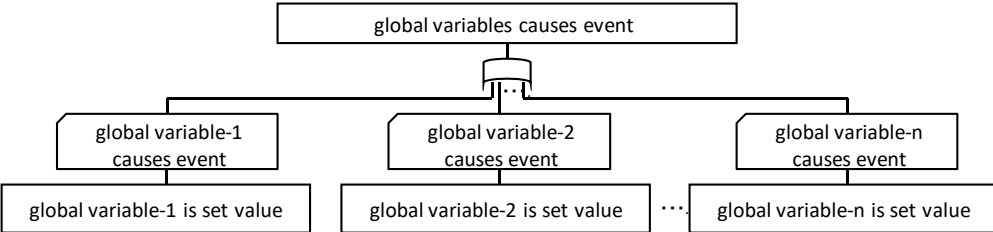

**Figure 9.** FTT for global variables.

### 3.2.7. FTT for Local Variables

Figure 10 shows the FTT for the local variables. Local variables exist within the relevant variable scope (instruction, function, etc.). This FTT expresses the fact that local variables cause events when there is a relationship with an instruction in which the local variable is set in the value immediately before.

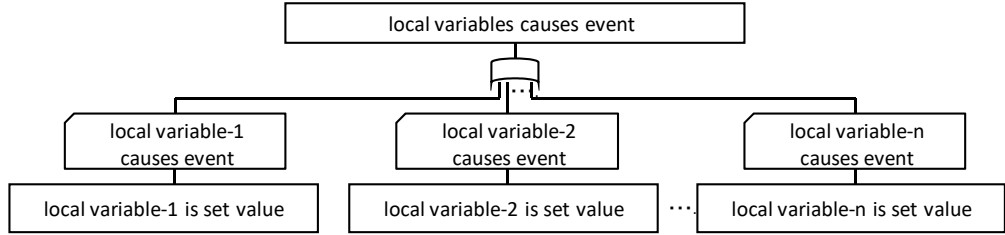

**Figure 10.** FTT for local variables.

### 3.2.8. FTT for Array

Figure 11 shows the FTT for an array. This FTT expresses the fact that "index of the array becomes less than 0," "the array causes the event when the N-th element does not exist (out of range, illegal index access)," or "the value stored in the N-th element is inappropriate."

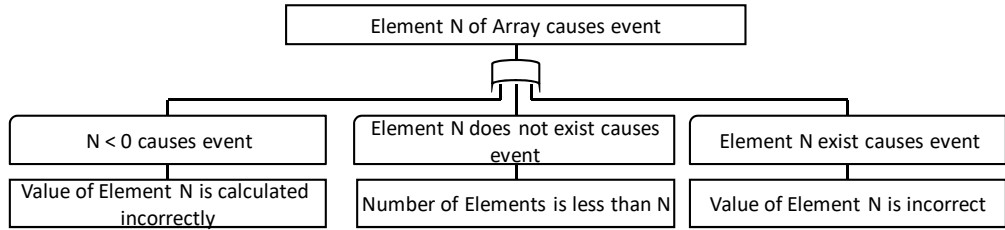

**Figure 11.** FTT for array.

### 3.2.9. FTT for Pointer

Figure 12 shows the FTT for a pointer. This FTT expresses the fact that the pointer causes an event when the address referenced by the pointer does not exist, the address is referenced by the other pointer, or the value stored in the address referenced by the pointer is inappropriate.

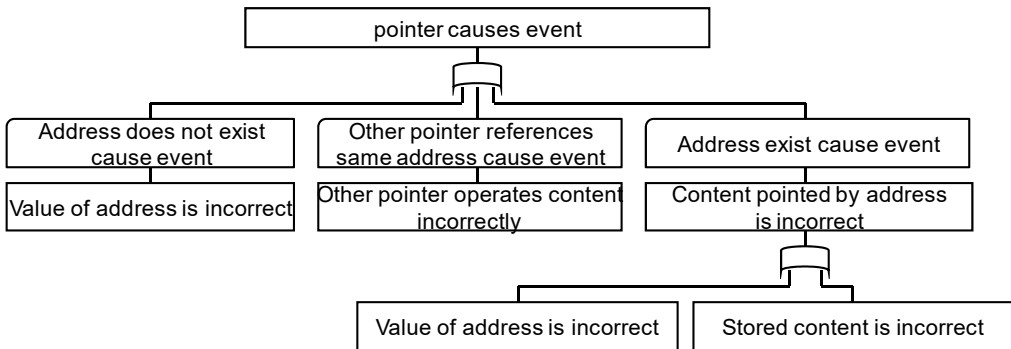

**Figure 12.** FTT for pointer.

### 3.2.10. FTT for Hierarchical Instruction

Here, we shall discuss an FTT when an instruction is hierarchically combined. Hereinafter, we refer to this kind of instruction as a hierarchical instruction. For example, an instruction where a tracked instruction exists within a controlled instruction, such as a block if instruction or while instruction, etc., is a hierarchical instruction. In this case, the controlled instruction including the tracked instruction is seen as one instruction, and to express that targeted instruction is executed when the conditions in the controlled instruction are established, FFTs are combined in the order of controlled instruction to targeted instruction. If the hierarchical instruction has three or more layers, the hierarchical instruction is seen as being within the scope of the furthest external instruction, and it connects the FTT from the furthest external instruction to the nearest internal instruction. At this time, the layer number for the furthest external layer is referred to as $L_{max}$, and the nearest internal hierarchical instruction is set to "1." The method of developing the FT for the hierarchical instruction is explained later under FT development rules in Section 3.3.

### 3.3. FT Development Rules

In this subsection, we discuss static program slicing (hereinafter, slicing) used to determine instructions executed just before the instruction being executed. Next, we discuss the ECSW information required for executing FTA. Third, we discuss FT development rules. Finally, we discuss FTA support tools.

### 3.3.1. Slicing

Slicing is a method of extracting all instructions that impact the calculation results for the variables focused on in the program, and the extracted instruction groups are called slices [24]. When the same input is applied to the original program and the slice, the calculation results for the variable that is being focused on shall be the same. A slice is a collection of all instructions with the control dependency on the focused instruction and with the data dependency on the variables included in the focused instruction. Here, a data dependency from $I_s$ to $I_t$ means that the variable values set in $I_s$ can be referenced with $I_t$. A control dependency from $I_s$ to $I_t$ means that $I_s$ is a branched instruction, and $I_t$ is included within that branch, or $I_s$ is a repeated instruction, and $I_t$ is included within the repetitions. The data dependency and control dependency from $I_s$ to $I_t$ are described respectively as $DD(I_s) \rightarrow I_t$, $CD(I_s) \rightarrow I_t$. Furthermore, $I_s$ and $I_t$ can be expressed as "instruction line number + target variable." Here, to obtain the instruction $I_n$ that generated <Event$_n$> from $I_1$ that generated <Event$_1$>, you need to follow the reverse-direction data dependency and reverse-direction control dependency. Instructions with a reverse-direction data dependency have the $I_s$ set for the value referenced by $I_t$ and the instructions with a reverse-direction control dependency have branches in which $I_t$ is included or a repetitive $I_s$. From here, these are described as $RDD(I_t) \rightarrow I_s$, and $RCD(I_t) \rightarrow I_s$, respectively. These can obtain the data dependency and control dependency, respectively. In general terms, the correspondences from $I_t$ to $I_s$ are one-to-many.

### 3.3.2. ECSW Information Required for FTA

Table 1 shows a list of information extracted from ECSW. This information is input into the FTA support tool.

**Table 1.** List of information extracted from embedded control software (ECSW).

| Information Name | Content |
|---|---|
| RDD list | Instructions with reverse-direction data dependency |
| RCD list | Instructions with reverse direction control dependency |
| Variable list | Variable name, type, valid scope, and substitution in the relevant variable |
| Substitution list | Substitute line, substitute expression (substituted variable name and operator) |
| Function list | Function name, return value type, function scope, dummy arguments, start-up type (cycle, interrupts), interrupt disabled timing, interrupt abled timing |
| Function call list | Called function name, calling position, argument |
| Instruction list | Instruction name, execution conditions, member block statement, nest number in member nest |
| Hierarchical instruction list | Instruction name, execution conditions, member statement, nest number maximum value |

### 3.3.3. FT Development Rules

Here, we explain the terms used in the FT Development Rules (FDR). The event at the top of each FTT (event immediately after executing the instruction) is referred to as the Top Event (TEvent), and the event at the bottom (event immediately before executing the instruction) is referred to as the Bottom Event (BEvent). The FTT has one TEvent and multiple BEvents. In addition, $\Phi$ expresses the FT initial state, and it only has one joint TEvent and BEvent. If Null is entered in the BEvent, this indicates that the event can no longer be tracked.

Figure 13 shows an overview of the FDR. The FT under development (dFT) refers to the FT being developed. After the first time that $\Phi$ is set for dFT, the FT setting is expressed with "$\rightarrow$". Next, we determine $<Event_1>$ and $I_1$, and set $<Event_1>$ for the dFT initial state $BEvent_1$. Here, $<Event_1>$ is a software fault resulting from the system-level FTA and $I_1$ is the instruction that causes the $<Event_1>$. Furthermore, FTs are developed until the content of all BEvents in dFT are Null. At this time, the FT is the completed version of the FT.

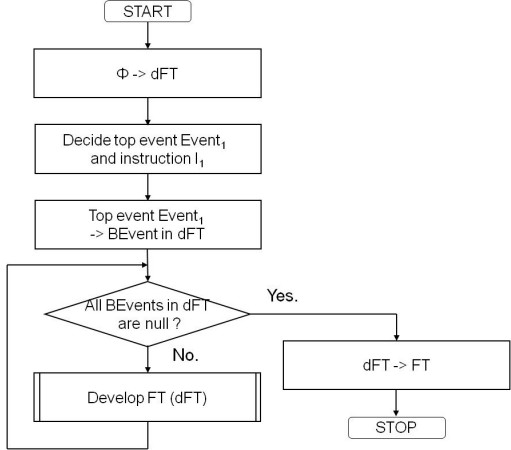

**Figure 13.** FT development rules -overview.

Figure 14 shows the development procedure for the FT under operation corresponding to the dFT BEvent (wFT) (section of dFT within Figure 14). Figure 14 describes the details in the module, while the development of FT (dFT) is described in Figure 13. wFT refers to the instructions included in the BEvent or the partial FT within the hierarchical instruction. First, in step 1 of Figure 14, $\Phi$ is substituted for wFT and initialized. Next, in step 2, the instructions included in the wFT BEvent are acquired, and a partial FT related to the instruction is developed. The development of the FT can be classified into cases where the instruction is a hierarchical instruction (RCD(I) not = $\varphi + \varphi$), as shown

in step 2.1, or is a non-hierarchical instruction (RDD(I) = $\varphi + \varphi$), as shown in step 2.2. Here, $\varphi + \varphi$ means the instruction line number and the target variable does not exist. In the case of a hierarchical instruction, the FTT of instruction for each layer is combined from the most outer layer (hierarchy $L_{max}$) to the most inner layer. Step 2.1.1 obtains the hierarchical instruction in which $I_g$ is included and calculates the $L_{max}$, which is the maximum layer of the hierarchical instruction and layer L where $I_g$ exists. In step 2.1.3, the FTT corresponding to $I_g$ is regarded as wFT. In step 2.1.4, BEvent in dFT is regarded as TEvent in wFT. Step 2.1.5 calculates all the BEvents in wFT. Step 2.1.6 combines the FTT for the global variable with wFT when the BEvent includes the global variables, and the FTT for local variables with wFT when the BEvent includes the local variables, and then wFT is combined with dFT. Finally, in step 2.1.7, this operation is repeated until it reaches a hierarchy in which the instruction being tracked exists. Next, the wFT is joined to the dFT to develop a new dFT. In the case it is not a hierarchical instruction, the FTT for this instruction is joined to the wFT, and, in a way, similar to the hierarchical instruction case, the FTT for global variables or local variables is combined with the wFT and, if necessary, the contents of the TEvent and the BEvents are modified. Then, the wFT is combined with the dFT to make a new dFT. Finally, in step 3, the developed dFT is returned.

### 3.4. FTA Support Tool

We prototyped an FTA support tool. For the development language, Java language and Apache POI were used. Apache POI is an API that is used to read and write Microsoft Excel format data in the Java application. As shown in Figure 2, the FTA support tool comprises the ECSW pre-processing tool, the information extraction tool, and the FTA support tool. The pre-processing tool converts ECSW instructions into instructions prepared for FFT. The information extraction tool extracts ECSW information shown in Table 1 from the replaced ECSW. The FTA support tool takes the replaced ECSW, the ECSW information, top event (<$Event_1$>), and the instructions generating this event ($I_1$) as input, and develops the FT. The FTA support tool shows the dFT draft to the analyst. The analyst confirms and modifies the content of the dFT's BEvent. The FTA support tool specifies the instruction executed immediately before based on the modified BEvent, develops a dFT by adding the FTT for this instruction, and shows the dFT to the analyst, and the analyst modifies the dFT. The analyst completes the FT by repeating this process. The FTA support tool creates the dFT in CSV format, which is then displayed using a spreadsheet application (Microsoft Excel). The reason the spreadsheet application is used is that the display style depends on the user favor. The user can change the displaying application easily. Additionally, as the dFT data are in CSV format, it can be exchanged easily between the FTA support tool, other displaying applications, and analysis tools.

---

**Process : Development of FT** : Develop FT corresponding to $BEvent_f (1 \leqq f \leqq m)$ in dFT.
1. $\varphi$ -> wFT.

2. Get $I_g$ that is included in $BEvent_f$ (1 <= f <= m) in the dFT, and develop a partial FT corresponding to $I_g$.
   2.1. When $RCD(I_g) \neq \varphi + \varphi$. (When $I_g$ is hierarchical instruction.)
     -- develop wFT corresponding to hierarchical instruction $I_g$ , and combine wFT to dFT --
   2.2. When $RCD(I_g) = \varphi + \varphi$. (When $I_m$ is not hierarchical instruction.)
     -- develop wFT corresponding to not hierarchical instruction $I_g$ , and combine wFT to dFT --

3. return dFT.

(**a**) FT development rules -details of FT developed in Figure 13 (1/5)—Outline of development of FT

**Figure 14.** *Cont*.

2.1. When $RCD(I_g) \neq \varphi + \varphi$. (When $I_g$ is hierarchical instruction.)

    2.1.1. Get hierarchical instruction belonging $I_g$, max value $L_{max}$ of layer of hierarchical instruction, and value L of layer that $I_g$ exist.

    2.1.2. $k = L_{max}$ .

    2.1.3. FT template for $I_g$ in k-th layer of hierarchical instruction -> wFT.

    2.1.4. $BEvent_f$ in dFT-> TEvent in wFT.

    2.1.5. Calculate $BEvent_i$ (1 <= i <= m) in wFT, and decribe it in $BEvent_i$.

    2.1.6. Conduct following sub-processes to all $BEvent_i$ (1 <= i <= m).

        2.1.6.1. When $BEvent_j$ in wFT include only global variables, conduct following sub-processes to all global variables.

            2.1.6.1.1. $BEvent_i$ in wFT  -> TEvent in FT template.

            2.1.6.1.2.  Calculated $BEvent_j$ (1 <= j <= n) in FT template for global variables, and describe it in $BEvent_j$ .

                  When $BEvent_j$ cannnot trace any more, null -> $BEvent_j$.

            2.1.6.1.3. wFT + FT template for global variables -> wFT.

            2.1.6.1.4. dFT + wFT -> dFT.

            2.1.6.1.5. $k = k - 1$.

        2.1.6.2. When $BEvent_j$ in wFT includes only local variables , conduct following sub-processes to all local variables.

            -- Conduct same sub-processes as 2.1.6.1 using FT template for local variables. --

        2.1.6.3. When $Bevent_i$ in wFT includes both global and local variables,  conduct same sub-processes as 2.1.6.1. and 2.1.6.2.

(**b**) FT development rules -details of FT developed in Figure 13 (2/5)—wFT development procedure when $I_g$ is hierarchical Instruction (detail of step 2.1).

    2.1.7. Develop FT for instruction in next layer.

        2.1.7.1. When $L < k < L_{max}$.

            2.1.7.1.1. FT template corresponding to $I_i$ in k-th layer of hierarchical instruction -> wFT.

            2.1.7.1.2. $BEvent_i$ in dFT -> TEvent in wFT. (Conducts those sub-processes to all i.)

            2.1.7.1.3. Calculate $BEvent_k$ (1 <= k <= o) in wFT, and describe it in $BEvent_k$.

            2.1.7.1.4. Conduct following sub-processes to all $BEvent_k$ (1 <= k <= o).

                2.1.7.1.4.1. When $BEvent_k$ in wFT includes only global variables, conduct following sub-processes to all global variables.

                2.1.7.1.4.1.1. $BEvent_k$ in wFT -> TEvent in FT template for global FT.

                2.1.7.1.4.1.2. Calculate $BEvent_l$ (1 <= l <=p) in FT template for global variables, and describe it in $BEvent_l$.

                    When $Bevent_l$ cannot trace any more, null -> $BEvent_l$.

                2.1.7.1.4.1.3. wFT + FT template for global variables -> wFT.

                2.1.7.1.4.1.4. dFT + wFT -> dFT.

                2.1.7.1.4.1.5. $k = k - 1$.

                2.1.7.1.4.1.6. goto 2.1.7.

              2.1.7.4.2. When $BEvent_k$ in wFT includes only local variables, conduct following sub-processes to all local variables.

                ---- Conduct same sub-processes as 3.1.4.7.1 using FT template for local variables. ----

              2.1.7.4.3.  When $BEvent_k$ in wFT includes global and local variables, conduct same sub-processes as 2.1.7.4.1. and 2.1.7.4.2.

(**c**) FT development rules -details of FT developed in Figure 13 (3/5)—wFT development procedure when $I_g$ is hierarchical Instruction (detail of step 2.1 cont')

**Figure 14.** *Cont.*

2.1.7.2. When k = L.
    2.1.7.2.1. FT template corresponding to $I_i$ in k-th layer of hierarchical instruction -> wFT.
    2.1.7.2.2. $BEvent_i$ in dFT-> TEvent in wFT. (Conduct those sub-processes to all i.)
    2.1.7.2.3. Calculated $BEvent_k$ (1 <= k <= o) in wFT, and describe it in $BEvent_k$.
    2.1.7.2.4. Conduct following sub-processes to all $BEvent_k$ (1 <= k <= o).
        2.1.7.2.4.1. When $BEvent_k$ in wFT includes only global variables, conduct following sub-processes to all global variables.
          2.1.7.2.4.1.1. $BEvent_k$ in wFT -> TEvent in the FT template for global variables.
          2.1.7.2.4.1.2. Calculated $BEvent_l$ (1 <= l <= p) in the FT template for global variables, and describe it in $BEvent_l$.
             When $BEvent_l$ cannot trace any more, null -> $BEvent_l$.
          2.1.7.2.4.1.3. wFT + FT_template for global variables -> wFT.
          2.1.7.2.4.1.4. developing _FT + wFT -> dFT.
          2.1.7.2.4.1.5. k = 0.
        2.1.7.2.4.2. When $BEvent_k$ in wFT includes only local variables, conduct following sub-processes to all local variables.
          -- Conduct same sub-processes as 2.1.7.2.4.1. using FT template for local variables. --
        2.1.7.2.4.3. When $BEvent_k$ in wFT includes global and local variables, conduct same sub-processes as 2.1.7.2.4.1. and 2.1.7.2.4.2.

(**d**) FT development rules -details of FT developed in Figure 13 (4/5)—wFT development procedure when $I_g$ is hierarchical Instruction (detail of step 2.1 cont').

2.2. When RCD($I_g$) = $\varphi$ + $\varphi$. (When $I_g$ is not hierarchical instruction.)
    2.2.1. Regard the FT template corresponding to $I_g$ as the wFT.
    2.2.2. $Event_g$ -> T_Event in the wFT.
    2.2.3. Describe result of calculated $BEvent_i$ (1 <= i <= m) in the wFT.
    2.2.4. Conduct following sub-processes to all $BEvent_i$ (1 <= i <= m).
        2.2.4.1. When $BEvent_j$ in the wFT includes only global variables, conduct following sub-processes to all global variables.
          2.2.4.1.1. $BEvent_j$ in wFT -> TEvent in the FT template for global variables.
          2.2.4.1.2. Calculate $BEvent_j$ (1 <= j <= n) in the FT template for global variables, and describe it in $BEvent_j$.
             When $BEvent_j$ cannot trace any more, null -> $BEvent_j$.
          2.2.4.1.3. wFT + FT template for global variables -> wFT.
          2.2.4.1.4. dFT + wFT -> dFT.
        2.2.4.2. When $BEvent_i$ in wFT includes only local variables, conduct following sub-processes to all local variables.
          -- Conduct same sub-processes as 2.2.4.1. using FT template for local variables. --
        2.2.4.3. When $BEvent_i$ in wFT includes global and local variables, conduct same sub-processes as both 2.2.4.1. and 2.2.4.2.

(**e**) FT development rules -details of FT developed in Figure 13 (5/5)—wFT development procedure when $I_g$ is not hierarchical Instruction (detail of step 2.2).

**Figure 14.** The development procedure for the FT.

## 4. Application and Evaluation

This section discusses an evaluation of the results of applying the proposed method to the existing ECSW and future issues.

*4.1. Application and Evaluation of the Proposed Method*

FTAs for "spin-stabilized satellite rotates too fast" written in [12] are conducted. The reason we conducted this application is that we felt there were no descriptions related to the middle analytic process that led to the top event when we read this result for the first time. We considered that the analyst omitted the middle analytic process based on his skills and experiences. We would like to confirm that the omitted parts can be complemented by applying the proposed method and the complete FT improves the understandability. Additionally, we apply and evaluate the proposed method for the existing five ECSWs.

4.1.1. Application and Evaluation of the Top Event for "Rotation Rate of the Satellite Became Too Fast"

Leveson et al. executed a system-level FTA related to an accident where the rotation rate of the spin-stabilized satellite became too fast, and the boom was torn off by centrifugal force. The result of this became an accident in cases where, in the top events, "value of variable period became too high" or "value of variable length became too low" in ECSW were caused. Figure 15 shows the ECSW structure chart. Vbrh and monitor spin are started with a timer interrupt, RESTART3 has an interrupt from the sun's pulses, and RESTART4 is started up with a clock interrupt. Figure 16 shows the source code rewritten in C (in [12], this is described in Pseudo Pascal).

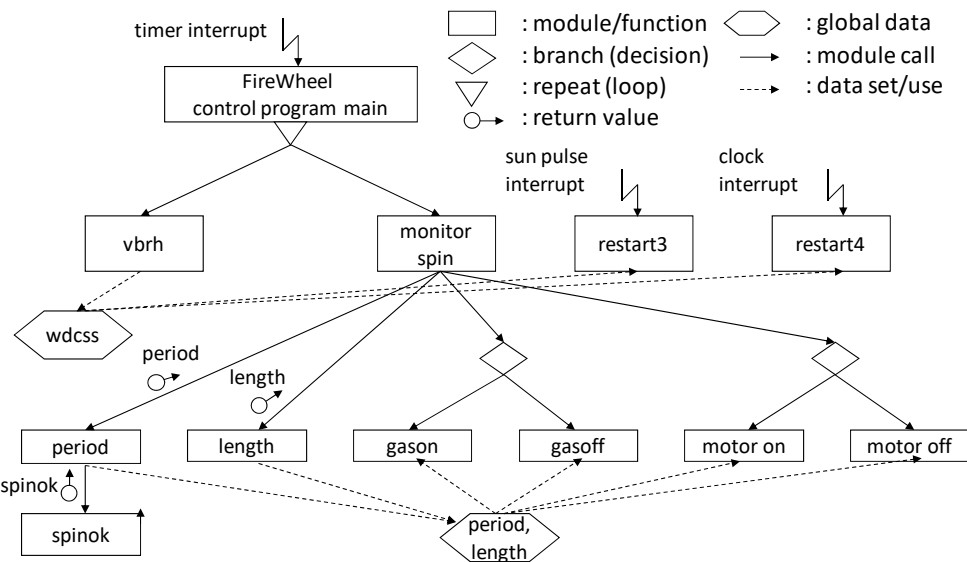

**Figure 15.** Outline of ECSW structure for spin stabilized satellite.

The number of events in the P_FT and L_FT is evaluated. The number of the event in L_FT is 20, while the number of the event in P_FTs is 56 (excluding events surrounded in the dotted line and supplementary comments). The reasons the P_FTs include many events are as follows.

- As for the P_FT, FDRs are strictly applied, and there is no omission of interim progress (in L_FT, the analyst omits the interim progress).
- As the module startup with interrupt is used many times, the FTT for interrupt is also used multiple times.
- As the global variables are used multiple times, the FTT for global variables are also used.
- In comparison to the FTT developed by Leveson, the FTT in the proposed method has many events.

We analyzed the correspondence between the two FTs. (I) to (VII) in Figures 17 and 18 express the section where the FTs correspond. The event group in P_ FT corresponding to each event in the L_FT is indicated by the section surrounded by a dotted line. The other group in Figure 18 is the section not described in Figure 17. From the above, we can see that, excluding Other, both FTs have the same

structure. In addition, we can see that the P_FT analyzes the cause of the fundamental events occurring in detail.

```
000: int SUNP , MAGP ;                          072: void RESTART3 ( ) {
002: int DNCTR , DNMAX , THETA ;                073:     SUNP = min ( LASTP , WDCSS ) ;
003: int WDCSS , WDCTR , LASTP, WDLOST, L1 , L2 ; 074:     int restkari ;
009:                                             075:     restkari = ( SUNP + 64 ) / 128 ;
010: int SAMPLE ( int SAMPLE_ARG ) {            076:     DNMAX = min ( restkari , 255 ) ;
011:     return SAMPLE_ARG ;                    077:     DNCTR = DNMAX ;
012: }                                          078:     THETA = 0 ;
013:                                             079:     LASTP = WDCSS ;
014: int PERIOD ( ) {                           080:     WDCSS = 0 ;
015:     int MS ;                               081: }
016:     int SUN , MAG ;                        103: void MONITORSPIN ( ) {
017:     bool ifc1 , ifc2 ;                     104:     int PERI ;
018:     ifc1 = SPINOK ( SUNP ) ;               105:     int LENG ;
019:     ifc2 = SPINOK ( MAGP ) ;               106:     PERI = PERIOD ( ) ;
020:     if ( ifc1 == true ) {                  107:     LENG = LENGTH ( ) ;
021:             MS = SUN ;                      108:     P = min ( AAA / 64 , 255 ) ;
022:     }                                      109:     L = min ( BBB / 16 , 15 ) ;
023:     else {                                 110:     if ( P < L ) {
024:             if ( ifc2 == true ) {          111:             GASOFF ( ) ;
025:                     MS = MAG ;              112:     }
026:             }                              113:     if ( P > L ) {
027:             else {                         114:             GASON ( ) ;
028:                     MS = SUN ;              115:     }
029:             }                              116: }
030:     }                                      117:
031:     if ( MS == SUN ) {                     118: int min ( int min_1 , int min_2 ) {
032:             return SUNP ;                   119:     return 0 ;
033:     }                                      120: }
034:     else {                                 121:
035:             return MAGP ;
036:     }
037:}
```

**Figure 16.** Source codes of spin-stabilized satellite -extracted-.

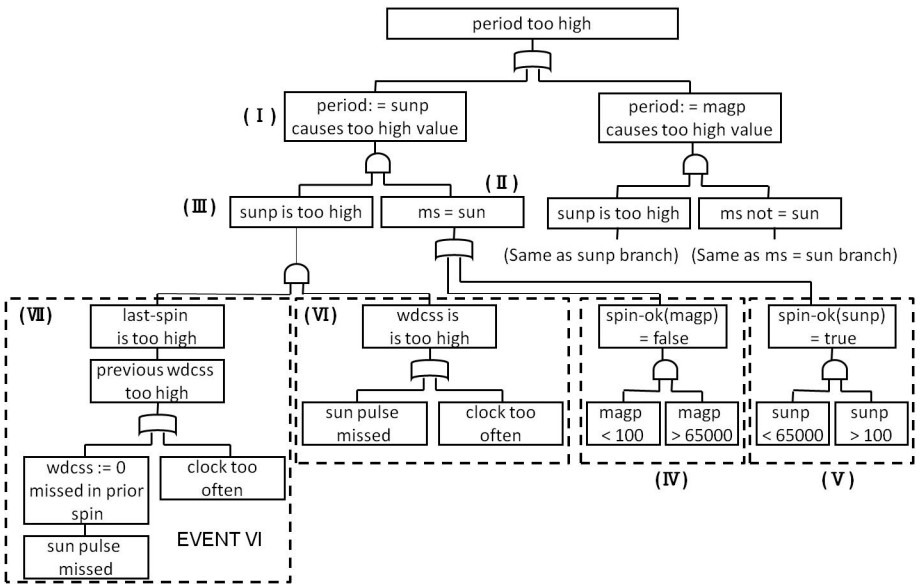

**Figure 17.** FT developed by Leveson.

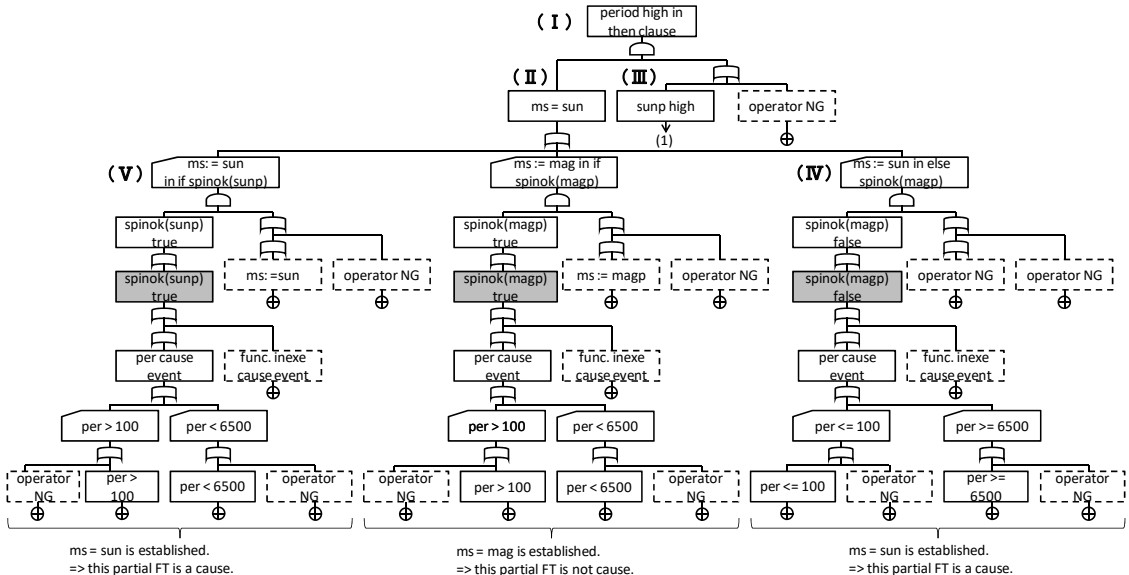

(**a**) FT developed using the proposed method.

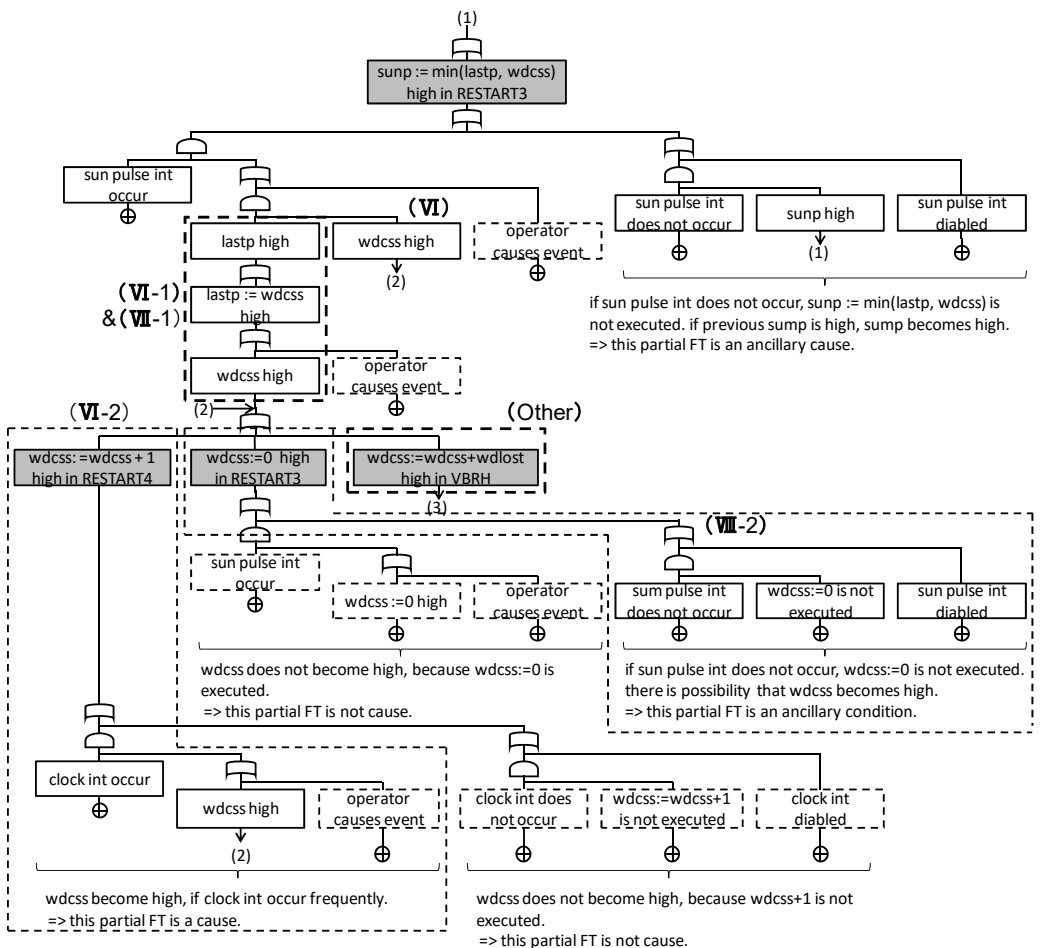

(**b**) FT developed using the proposed method.

**Figure 18.** *Cont.*

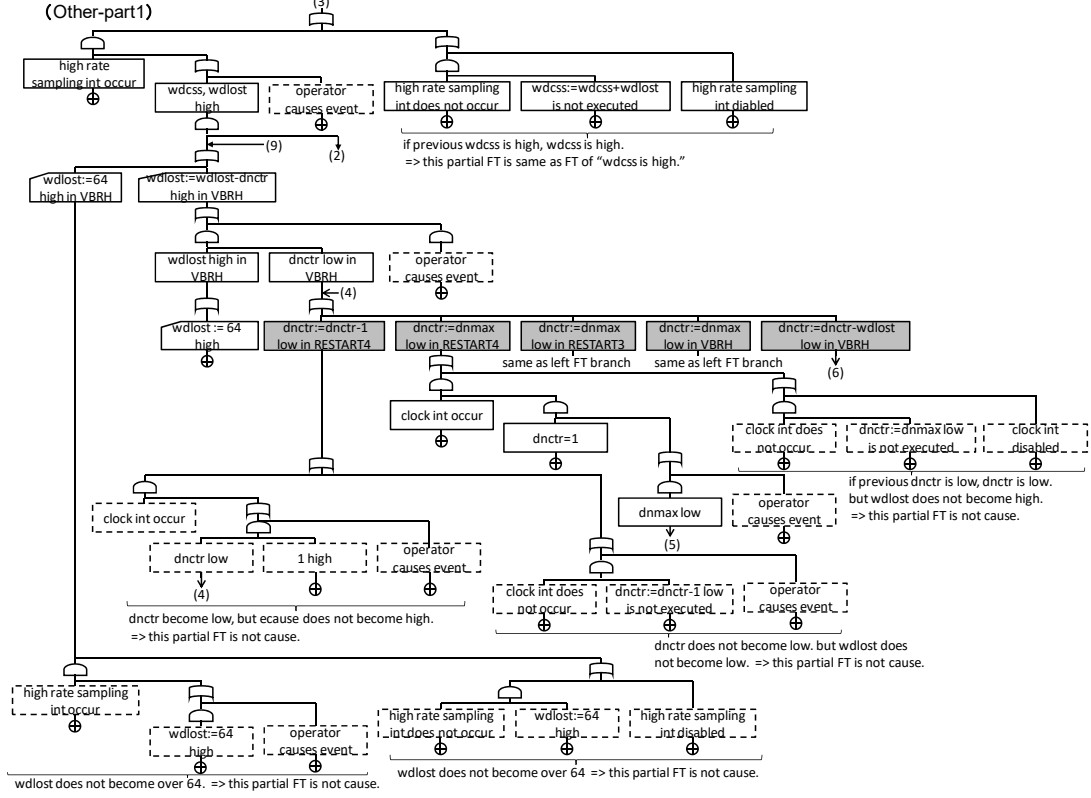

(**c**) FT developed using the proposed method.

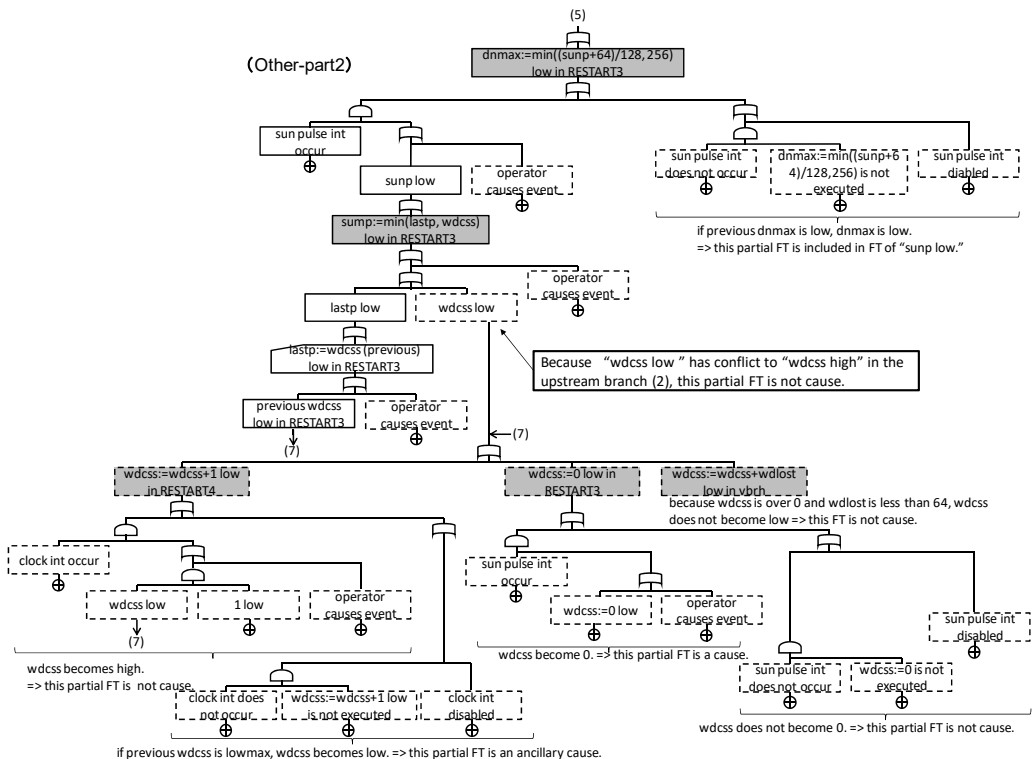

(**d**) FT developed using the proposed method.

**Figure 18.** FT developed using the proposed method.

In the case of this top event, the total time for conducting FTA takes about 3.0 h. The details of the FTA time are 0.5 h for understanding the source code (this task is conducted manually) and 2.5 h for developing FTAs (this task is conducted automatically using the developed tools). Here, the developing time of L_FT is unknown. The time is appropriate and acceptable for developing P_FTs. The reason the proposed method takes a short time for developing FTs is that the proposed method takes a shorter time to search the previously executed instructions.

Finally, we review the detected fundamental events. With L_FT, "magp < 100 and magp >" or "sunp > 100 and sunp <" are established, and "missing the sun pulse" or "the clock becoming too fast" were fundamental events. On the other hand, with P_FT, based on (V) and (IV) in Figure 18a, if "magp < 100 and magp >" or "sunp > 100 and sunp <" are established, then, from (VI) in Figure 18b, "the clock interrupt occurs frequently," or from (VII) in Figure 18b, "sun pulse interrupts do not occur" are the fundamental events. Although the description is different, "missing the sun pulse" and "sun pulse interrupt does not occur" are similar content.

Here, we describe the FTA result for the top event "value of variable length is too low." The evaluation method is the same as the former example. There were 15 L_FT events and 54 P_FT events. The two FTs had a similar structure. With L_FT, the fundamental events were "clock does not occur," "interrupt prohibition was not released," and "return value of function sample is too small." With P_ FT, on the other hand, "clock is not generated," "interrupt prohibition is not released", and "return value of function sample was inappropriate." These have the same content, and the reason there is a large number of P_FT events is the same as for the previous results. The time for conducting FTA takes about 2.0 h. The details of the FTA time are 0.5 h for understanding the source code (this task is conducted manually) and 1.5 h for developing FTAs (this task is conducted automatically using the developed tools). Here, the developing time of L_FT is unknown. The time is appropriate and acceptable for developing P_FTs. The reason the proposed method develops P_FTs within the appropriate and acceptable time is the same as in the former example.

From the above results, we can see that appropriate FTs can be developed corresponding to the top events in the proposed method within the appropriate time. As the number of the middle event in P_FTs becomes three times the number of the middle event in L_FT, there exists a concern about understanding the outline of the P_FTs. Therefore, P_FTs are shown as the hierarchical FTs in Figure 19. This FT is easily developed from P_FT by summarizing the middle events as a group. Figure 19 is the top-level FT, and lower-level FTs are the same as the FTs shown in Figure 18b–d. As a result, the analyst insists that displaying FT as the hierarchical FT can improve the readability. Additionally, an analyst insists that the analyzing process in the P_FTs is understandable because the P_FTs do not omit the middle process of conducting FTA.

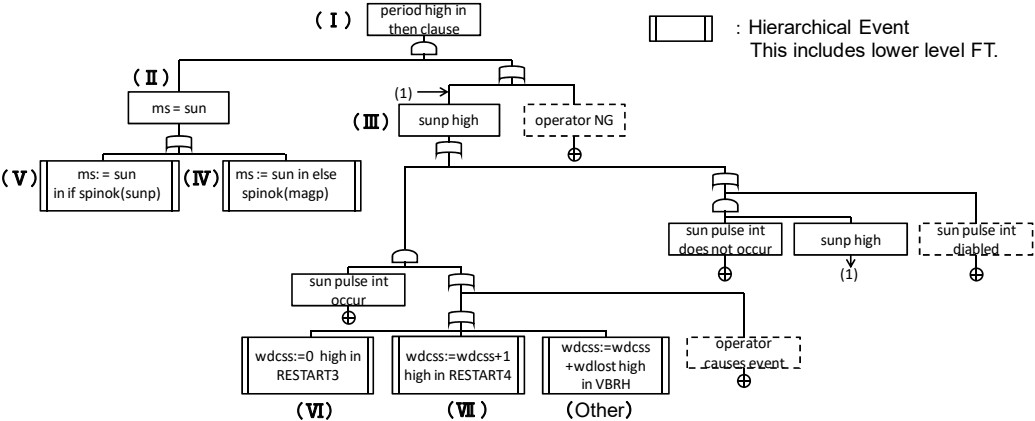

**Figure 19.** Top-level FT shown in hierarchical format.

### 4.1.2. Application and Evaluation Applying to Existing ECSW

In order to be evaluated, the proposed method is applied for the five existing ECSWs. Those are the ECSWs that control the electrical device, and the size of the ECSW is 30–200 LOC written in the C language. Two analysts who have over three year's experience and the same skills conduct FTA individually. One analyst develops the FT based on his experiences (hereafter, E_FT), and the other analyst develops the FT using the proposed method (hereafter, P_FT). After developing the FT, analysts evaluate the FTs with each other. Table 2 shows the outline of the ECSWs, top event, and the size of the ECSW. Table 3 shows the evaluation results of P_FT and E_FT, such as the fundamental events, number of events, developing time using the developed tools (except for the time for understanding the code), and the validity of the FT. As a result, in Table 3, in example No.5, P_FTs are appropriate, while E_FTs miss the fundamental event where the value in the buffer is too low. This result shows a possibility that applying the proposed method reduces the oversight of the fundamental events. The number of events in P_FTs is two times larger than the number of events in E_FTs. The considered reason is that the number of used interrupts and the number of the used global variables are small in comparison to the case described in Section 4.1.1. The FT developing time of P_FTs and E_FTs is the same. However, in the case of analyzing the ECSW with complex structure, it is considered that the analyzing time for P_FT becomes longer because the proposed method has to analyze the FT branches, which could not be the fundamental events. This issue can be resolved by excluding such branches. This is the trade-off between the FT developing time and oversight of fundamental events.

**Table 2.** Results of application of the proposed method.

| No | Outline of ECSW | Top Event | LOC |
|----|-----------------|-----------|-----|
| 1 | Data differing from the cycle are displayed on the screen. | Display value is too large. | 33 |
| 2 | Hours, minutes, and seconds of passing time are displayed alternately. | Passing time is calculated incorrectly. | 55 |
| 3 | Input the motor operation time, and rotate the motor for that time. | Motor does not stop. | 81 |
| 4 | The current time of flashing on the LED. | The time is slow. | 207 |
| 5 | Execute an instruction corresponding to a three-digit number. | A specific instruction cannot be executed. | 115 |

**Table 3.** Summary of the FT evaluation.

| No | Fundamental Events | P_FT | | | E_FT | | |
|----|--------------------|------|-----|--------|------|-----|--------|
| | | Num. of Events | Hours | Validity | Num. of Events | Hours | Validity |
| 1 | Inappropriate value of the counter, and 2 others | 18 | 0.5 | App [1] | 10 | 0.5 | App [1] |
| 2 | Not to occur timer interrupt, and 2 others | 15 | 0.5 | App | 7 | 0.5 | App |
| 3 | Stop switch OFF, and 2 others | 15 | 0.5 | App | 12 | 0.5 | App |
| 4 | Not to occur interrupt, and the false value of judgment flag | 26 | 2 | App | 15 | 1 | App |
| 5 | The large value of the buffer, or small value of the buffer | 45 | 2 | App | 15 | 1.5 | Ovs [2] |

[1] App means that the result of FT is appropriate. [2] Ovs means that the result of FT has some oversights.

### 4.2. Issues in the Proposed Method

This subsection describes the issues in the proposed method.

### 4.2.1. Issues Related to the Scale of ECSW

The size of the ECSW used for the evaluation was 50–200 LOC. In recent years, the scale of the ESCW has grown to realize high performance. In addition, with the ECSW, data sending and receiving using global variables are often used, and module startup using multiple interrupts is used. Therefore, the structure of the developed FT using the proposed method is considered to be complex, and it is feared that the level of understanding will be low and the analysis time will be long. It is necessary to introduce the design standard, such as ISO26262 [6], etc., that requires the usage restrictions on global variables and the multiple interrupts.

### 4.2.2. Issues Related to the ECSW's Dirty Structure

There are many ECSWs that are not applying the appropriate design process and standard. In many cases, those ECSWs have dirty structure (structure without appropriate functional dividing and design). When applying the proposed method to those ECSWs, it becomes highly possible that the developed FTs become more complex and larger. We investigate the refactoring method for the ECSWs with dirty structure before conducting FTA.

### 4.2.3. Issues Related to the Large Number of FT Events

When developing the FT using the proposed method, the number of the event in the FT becomes larger. While this can realize the development without the oversights of information related to the FT, the readability of the FT is decreased. To resolve this problem, we are going to develop the tools that show the FT with hierarchical expression (like Figure 19) and prunes the FT branch using Boolean algebra.

### 4.2.4. Issues Related to the Judgment of the Analyst

When developing the FT, the same section of the FT within the FT may be repeated multiple times. This type of FT structure occurs where the tracked instruction exists within the repetition instruction or interrupt processing occurs frequently. In this kind of FT structure, there are two types, such as (1) the top event occurs when the process is repeated until the certain times, and (2) the event goes into an infinite loop. In the proposed method, there is no judgment made as to which stage to stop the analysis. Therefore, we add a function that notifies the information that the repetitions reach a certain number. By using this information, the analyst can decide whether to continue or stop the analysis.

### 4.2.5. Issues Related to Object-Oriented Languages

If we define FTT for instructions in other procedural programming languages, the proposed method can be applied. On the other hand, the proposed method cannot be applied to ECSW written in object-oriented languages. This is because the FTTs and FDR in the proposed method give no consideration to inheritance and polymorphism that are features of object-oriented programming. We would like to investigate the proposed method for object-oriented programming in the future.

## 5. Summary

In this paper, we define FTT and FDR, and propose an FTA method. As a result of applying the proposed method to the top event of the existing ECSWs, we could confirm that appropriate FTs were developed, and appropriate fundamental events were detected. These show the effectiveness of the proposed method and support tool. In the proposed method, as we developed the FT complying with the application of the FTTs and FDR, the quality of FT information improved. Additionally, we improved the level of understanding within the appropriate time. In the future, we shall apply the proposed method to large-scale ECSW, and feed-back the results into the proposed method.

**Author Contributions:** Conceptualization, M.T.; Discussion, M.T., Y.A. and Y.W.; writing—original draft preparation, M.T.; writing—review and editing, M.T., Y.A. and Y.W.; supervision, M.T.; funding acquisition, M.T.; All authors have read and agreed to the published version of the manuscript.

**Funding:** This research was supported by a Grant-in-Aid for Scientific Research (C) of the Japan Society for the Promotion of Science, grant number 19K04920, title "Integrated analysis method for hazard caused by software interaction cooperating with multiple safety analysis methods".

**Conflicts of Interest:** The authors declare no conflict of interest.

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
