# Peer review of "A Proposal of Fault Tree Analysis for Embedded Control Software"

_information, doi:10.3390/info11090402_

Round 1

Reviewer 1 Report

The revised paper has incorporated the feedback from reviewers.

Reviewer 2 Report

I am satisfied with paper improvements, below I add two additional suggestions:

Line 86: I suggest to replace dot after word “is used” by comma as follows:”…. issued, details…”

Line 589: I suggest to change “By using the information,…” into “By using this information,…”

This manuscript is a resubmission of an earlier submission. The following is a list of the peer review reports and author responses from that submission.

Round 1

Reviewer 1 Report

This paper presents in interesting and relevant topic, namely the creation of Fault Trees (from a template) for embedded control software.

Unfortunately, I find the paper hard to read, and the methodology is insufficiently clear. Hence I cannot recommend acceptance.

Also, the case study that is performed is an incredibly simple programmes from the '80. This does not generalize to today's complex systems. 

  • The abstract claims that an analist would always obtain an appropriate results of FTA, no matter what skills. I do not believe this: the quality of a model always depends on the skills, also for FTA
  • Introduction: Why is it important which language the software is written in?
  • Introduction: it would be good to also introduce the case study + results already here
  • Related work: Related work is decent, except that there has been extensive related work on Software Fault Trees. That is not covered here.
  • Sect 3: here I have troubles following the authors.
    • The ECSW operates normally until a certain instruction. How is that instruction determined? 
    • Unclear what the role of the execution is
    • What are Event_101, etc? 
  • Sect 3.2: do I need a FT element for every instruction, block, statement, global variable?? How is this going to scale?
  • Fig 15: Difficult to read
  • Fig 18, 18a and further: also difficult to read. 

Author Response

Thank you for your important suggestions and comments. Authors revised this paper according to your suggestions and comments.

  • Comment 1

The proposed method is applied to 80’s system in Case Study

  • Answer to Comment 1

authors add the reason in the beginning of 4.1 why the proposed method was applied to the system in the 1980s. In recent years, embedded systems have complicated hardware and software. However, simple embedded systems still make up a high rate (described in the introduction). The cost (time, engineers, etc.) that can be spent is limited for the developments of small-scaled embedded system. The proposed method aims at helping to conduct safety analysis for such small-scaled system at low cost. After establishing the proposed method, we plan to investigate the safety analysis for complex ECSW as you pointed out. Thank you for your valuable comments.

  • Comment 2
  • Clarify the skills of the analyst.
  • The description of analyst’s skill is added to the abstract. Analysts, who have conducted application cases, have been studying information science and safety engineering for a few years.

  • Comment 3
  • Why is it important which language written in?
  • Answer to Comment 4
  • The reason is added in Section1 (line 33 in information-803281_re-submit_for review process (modifications are shown).doc file).

  • Comment 4

More related works should be added.

  • Answer to Comment 4

Research papers related to the application of FTA [20-23] are added to the references.

  • Comment 4

How is that instruction determined?

  • Answer to Comment 4

The method to decide the target instruction of FTA is added to 3.1.1 (Line 153 in information-803281_re-submit_for review process(modifications are shown).doc file).

  • Comment 5

What is Event_101, etc?

  • Answer to comment 5

Event 101 etc. represent the event that occurred as a result of execution of the instruction immediately before.

  • Comment 6

How is FT element going to scale?

  • Answer to 6

As the authors describe in 4.2.3, 4.2.4, FTA becomes difficult using the proposed method when applying the ECSW with duty structure and large scale. It will take long time and difficult to analyze the developed FT, if FT can be developed. The authors should obey the development standard (such as, adequate module size, no-use of global variables size module). This problem is our future work. Thank you for your important suggestion.   

  • Comment 7

Figure 15, difficult to read.

  • Answer to Comment 7

The expressions in Fig. 15 complies with the standard description method of the structure chart proposed by M. Page-Jones et al. (Some functions such as interrupt signals have been added). As the authors consider that it is readable, the authors have not corrected it.

  • Comment 8

Figure 18, difficult to read.

  • Answer to Comment 8

As for Figure 18, (2) and (4) have been expanded. However, due to the page width limitation, (1) and (3) could not be expanded. It is considered that these figures are easier to see if they are displayed on a large size monitor (in the case of printing, it is easier to see if printed with A3).

Reviewer 2 Report

The topic of the paper seems to attract potential readers due to an important issue of dependability issues in embedded systems. Unfortunately, the original contribution and its practical significance is not stated clearly in the introduction, what is the input and output of your analysis. Then this should be followed by an outline of the paper (its organization and logical setup). Section 2 should be extended and referenced to more publications related to the topic. The list of references includes 8 positions relevant to some standards which in fact are rather loosely related to the main topic of the paper. It is not clear what is a new contribution (or extension) in relevance to previous author papers [9, 14, 19], there is some imbalance of newer publications within the remaining positions, some of them are quite old. More newer ones related to FT or relevant techniques applied to embedded systems are needed, can the FT technique be combined with fault injection  or complement it?. For example I enclose a few of them:

  1. Kloos, T. Hussain, R. Eschbach, Risk-Based Testing of Safety-Critical Embedded Systems Driven by Fault Tree Analysis, 2011 IEEE Fourth International Conference on Software Testing, Verification and Validation Workshops,
  2. Chen, et al., Systems Modeling with EAST-ADL for Fault Tree Analysis through HiP-HOPS,  IFAC Proceedings Volumes, Volume 46, Issue 22, 2013, Pages 91-96
  3. Trawczyński D., Sosnowski J., Gawkowski P. (2010) Testing Distributed ABS System with Fault Injection. In: Sobh T., Elleithy K. (eds) Innovations in Computing Sciences and Software Engineering. Springer, pp. 201-206
  4. Dabboussi, R. Kouta, J. Gaber, M. Wack, B. E. Hassan and L. Nachabeh, "Fault tree analysis for the intelligent vehicular networks," 2018 IEEE Middle East and North Africa Communications Conference (MENACOMM), 2018, pp. 1-6,

Section 3 lacks clear problem statement: what kind of faults do you take into account, how they will be identified, what will result from applying your analysis (identification of program imperfections, suggestions of improvements, mitigation of faults, evaluation of relevant losses, etc. ?? More substantive presentation of your method in relevance to fig, 2 is needed. Figures 3-12 are very simple and practically redundant to the explanation text , moreover you refer to a very old paper [12]. The saved space could be used for describing in a more detailed way sections 3.3 and 3.4. Fig. 14 needs extended explanation in the text. Section 4. 1 is an illustration of applying your method to a simple problem from [12], it is not clear what do you achieve here as compared with [12], by the way why do you use so old example?? Better comments are needed to follow the included figures, show the reader what is important in these figures? Nowadays hardware and software of embedded systems is quite complex, how your method (and the tool) scales for more complex systems, what are the limitations?? The paper summary is too superficial and loosely correlated with the paper sections. Improving the paper it is also reasonable to make it more attractive to readers partially or not familiar with FT. 

In some points English needs improvement, e.g. page 3, line 106, sentence: “We describe the process which top events occur in ECSW”

Author Response

Thank you for your important suggestions and comments. Authors revised this paper according to your suggestions and comments.

  • Comment 1

Add original contributions

  • Answer to Comment 1.

The description of our contribution is added to section 1.

  • Comment 2

What is input and output of the proposed method?

  • Answer to Comment 2

The input and output of the proposed method are added to section 1. (line86 in information-803281_re-submit_for review process (modifications are shown).doc file)

  • Comment 3

Add outline of this paper

  • Answer to Comment 3

The organization of this paper is added to last part in Section 1. (line90 in information-803281_re-submit_for review process (modifications are shown).doc file)

  • Comment 4

Add more publications.

  • Answer to Comment 4

Research papers related to the application of FTA [20-23] are added to the references.

  • Comment 5

What kind of the fault do you take into account?

  • Answer to Comment 5

The description related to the fault targeted to this research and its identification method is added to the beginning part in 3.1.1. (line153 in information-803281_re-submit_for review process (modifications are shown).doc file)

  • Comment 6
  • Description of Figure 3-12 is duplicated (same explanation exist in the paper).
  • Answer to Comment 6
  • There is no limitation about pages of the paper. To help the understanding this paper, the authors did not delete the description of Figure 3 to 12.

  • Comment 7

Figure 14 is difficult to read.

  • Answer to Comment 7

To improve the readability of Figure 14, this figure is divided into plural figures (the figure that shows outline of the procedure, the two figures that describe the developing procedure of the hierarchical instruction, the figure that shows the developing procedure of non-hierarchical instruction), and the font size change larger.

  • Comment 8

Why do you use old example?

  • Answer to Comment 8

The authors add the reason in the beginning of 4.1 why the proposed method was applied to the system in the 1980s. In recent years, embedded systems have complicated hardware and software. However, simple embedded systems still make up a high rate (described in the introduction). The cost (time, engineers, etc.) that can be spent is limited for the developments of small-scaled embedded system. The proposed method aims at helping to conduct safety analysis for such small-scaled system at low cost. After establishing the proposed method, we plan to invetigate the safety analysis for complex ECSW as you pointed out.

  • Comment 9

Better comments are needed to follow the included figure.

  • Answer to Comment 9

Because the authors consider that Figure 15 and 16 is necessary to show the outline of application, the authors do not delete those figures.

  • Comment 10

Nowadays, hardware and software of embedded system become complex. How do your method scales for more complex system?

  • Answer to Comment 10

As you point out, hardware and software are complicatedly configured in large-scaled embedded systems. However, simple embedded systems still make up a high rate (described in the introduction). The cost (time, engineers, etc.) that can be spent is limited for the developments of small-scaled embedded system. The proposed method aims at helping to conduct safety analysis for such small-scaled system at low cost. After establishing the proposed method, we plan to investigate the safety analysis for complex ECSW as you pointed out. Thank you for your valuable comments.

Reviewer 3 Report

The authors should add the background of the study and why the proposed method is better compared to the existing approaches. A case study to illustrate it is helpful. Research methodology can be clarified better. The authors should highlight the advantages and disadvantages of the proposed method compare to the existing ones.

Author Response

Thank you for your important suggestions and comments. Authors revised this paper according to your suggestions and comments.

  • Comment 1
  • Add the background this study.
  • Answer to comment 1
  • The authors add the description of the background of this study to the section 1.

  • Comment 2

Add superiority (advantage) of this research.

  • Answer to Comment 2

The description of the superiority of the proposed method is added to the section 4.1.

  • Comment 3

Add disadvantage of this research

  • Answer to Comment 4

The description of disadvantages (issues, the problems which have been not resolved) of this research are described in the section 4.2. As there are some inaccurate description in the last part in the section 4.2.4, the authors revise the description.

Reviewer 4 Report

1.

Grammar needs many corrections which are annotated in the attached pdf file.

2.

Sec. 3.2 contains the subsections 1, 2, 3, ..., 10. Such numbering may be confused with the numbering of the main sections (1. Introduction, 2. Related work, ...). I suggest the replacement of 1, 2, 3, ..., 10 with A, B, C, ... or with Roman numbers I, II, III, ..., X.

3.

In sec. 4.1.1, at lines 394, 395, 413, 414, the authors talk about the hours necessary to prepare the fault tree (FT) models. Does it mean that the proposed method must be manually applied? Can it be automated by means of software tools? In fig. 2 the outline of the method is shown and contains software components like databases and ECSW analysis tool. This point must be clarified in the paper.

4.

In sec. 4.1.2, at line 444, there is a reference to 4.1.2, but it is not clear what 4.1.2 refers to (section? table? bibliography?). I believe that it must be corrected.

5.

In sec. 4.2.4, at line 487, the sentence "has them make a judgment" is unclear and must be rephrased.

6.

Tab. 2 is deployed on pages 17 and 18. It would be more readable if it appeared in only one page.

Author Response

Thank you for your important suggestions and comments. Authors revised this paper according to your suggestions and comments.

  • Comment 1

Referring to the attached file, the author modifies pointed parts.

  • Answer to Comment 1

Please check the new Word file, the modified parts are high-lighted using word

  • Comment 2

Change subsection number 1-10 in Section 3.2

  • Answer to Comment 2

The numbers of subsection 1-10 are replaced into the symbol of I-X

  • Comment 3

Clarify manual operation and automation in 4.1.1.

  • Answer to Comment 3

The description of the tasks conducted manually and automatically (conducted using FTA support tools) are added in the 4.1.1.

  • Comment 4

Reference to 4.1.2 in line 444 is incorrect.

  • Answer to Comment 4

“Section 4.1.2” at line 444 is mistake of “Section 4.1.1”. The authors modify this section number. Please show the file named “information-803281_re-submit_for review process (modifications are shown).doc” in line 555.

  • Comment 5
  • The sentence “has them make a judgement” is unclear.
  • Answer to Comment 5

As this sentence is difficult to understand, the authors modify it.

“By using the information, the analyst can decide whether to continue or stop analysis.” Please show the file named “information-803281_re-submit_for review process (modifications are shown).doc” in line 605.

  • Comment 6

Table 2 is deployed on page 17 and 18.

  • Answer to Comment 6

The authors modified Table 2 to fit on one page.

Round 2

Reviewer 1 Report

This paper describes a relevant topic. However, I recommend to reject this paper. 

The problem that I have with the paper is that (1) I still do not understand how this method really works. This needs a major rewrite of the paper

Furthermore, (2) experimental evaluation is insufficient (3) related work is still insufficient.  (4) I am concerned with scalability

(1) The introduction states that the system level FT clarifies the system fault, amd they clarify the hardware and software faults and parts (instruction of ECSW) that cause a system-level fault FTA  executed for the software fault. First, we will discuss top events (Faults).

  • What does it mean to "clarify a fault"?
  • The system level FT is still created by an expert, so still depending on the skill of the analyst
  • I have troubles understanding section 3.1.1. The authors assume that each instruction is succeeded by an event. This assumes a synchronous computation paradigm. 
  • FIg 1 is also unclear: what are the branches, what are the interrups? 
  • It would really help to present a concrete example, with concrete code and concrete faults and failures.
  • Figure 3 is not clear to me either: how are the various results and templates combined in the FTA tool?

(2) The method is evaluated on 5 systems. The LoC is very low. First the tables 2 and 3 are not properly explained; it is all described quite imprecise and sloppy. I like to know

  • what criterion is evaluated? why these?
  • I like to see the size of the FT, size of the system, analysis times etc. 
  • Where do those systems come from? What application type?

(3) In terms of related work, I expect work on software fault trees to be included here. This is quite extensive.

(4) if there is a FT element for each instruction / LoC in the system, I do not see how this method can ever scale. 

The additional text in the new version did not help me to better understand this paper. Therefore, unfortunately, I still recommend to not accept this paper.